# Transcription factor EB (TFEB) activity increases resistance of TNBC stem cells to metabolic stress

Milad Soleimani[1,2,*], Mark Duchow[2,*], Ria Goyal[2] ⓘ, Alexander Somma[2], Tamer S Kaoud[3], Kevin N Dalby[1,3], Jeanne Kowalski[2] ⓘ, S Gail Eckhardt[1,2], Carla Van Den Berg[1,2,4] ⓘ

Breast cancer stem cells (CSCs) are difficult to therapeutically target, but continued efforts are critical given their contribution to tumor heterogeneity and treatment resistance in triple-negative breast cancer. CSC properties are influenced by metabolic stress, but specific mechanisms are lacking for effective drug intervention. Our previous work on TFEB suggested a key function in CSC metabolism. Indeed, TFEB knockdown (KD) inhibited mammosphere formation in vitro and tumor initiation/growth in vivo. These phenotypic effects were accompanied by a decline in CD44[high]/CD24[low] cells. Glycolysis inhibitor 2-deoxy-D-glucose (2-DG) induced TFEB nuclear translocation, indicative of TFEB transcriptional activity. TFEB KD blunted, whereas TFEB (S142A) augmented 2-DG–driven unfolded protein response (UPR) mediators, notably BiP/HSPA5 and CHOP. Like TFEB KD, silencing BiP/HSPA5 inhibited CSC self-renewal, suggesting that TFEB augments UPR-related survival. Further studies showed that TFEB KD attenuated 2-DG–directed autophagy, suggesting a mechanism whereby TFEB protects CSCs against 2-DG–induced stress. Our data indicate that TFEB modulates CSC metabolic stress response via autophagy and UPR. These findings reveal the novel role of TFEB in regulating CSCs during metabolic stress in triple-negative breast cancer.

## Introduction

Triple-negative breast cancer (TNBC) is a breast cancer subtype defined by the absence of progesterone receptor (PR), estrogen receptor (ER), and human epidermal growth factor receptor 2 (HER2) (Won & Spruck, 2020). TNBC tumors are characterized by a high degree of heterogeneity, early metastatic onset, treatment resistance, and tumor relapse. Treatment of TNBC is more challenging than other breast cancer subtypes because of a lack of well-defined therapeutic targets and high tumor heterogeneity, leading to poor patient prognosis (Vagia et al, 2020; Bai et al, 2021).

Breast cancer stem cells (CSCs) are a subpopulation of mostly quiescent cells that exhibit the capacity to self-renew and differentiate to reconstitute heterogeneous cell populations reflecting those of the original tumor. Mounting evidence points to TNBC harboring more CSCs than other breast cancer subtypes (Honeth et al, 2008; Ma et al, 2014). Indeed, CSCs contribute to the characteristic tumor heterogeneity, frequent metastasis, treatment resistance, and disease relapse observed with TNBC (Marra et al, 2020). This relatively small subpopulation is unique in terms of metabolic needs and plasticity, capable of adapting to metabolic and oxidative stress (Snyder et al, 2018). Understanding the metabolic mechanisms that govern CSC character and persistence could reveal new therapeutic avenues for TNBC.

The lysosomal membrane serves as a hub for metabolic signaling as lysosomes collect damaged organelles and recycle nutrients. Transcription factor EB (TFEB) is a basic helix–loop–helix, leucine zipper transcription factor best known for its key role in regulating lysosomal biogenesis and autophagy (Napolitano & Ballabio, 2016). Along with melanocyte-inducing transcription factor (MITF), TFEC, and TFE3, it belongs to the microphthalmia/transcription factor E (MiT/TFE) family of transcription factors (Tan et al, 2022). TFEB phosphorylation, to a large extent, determines TFEB subcellular localization and activity. Once unphosphorylated, TFEB localizes to the nucleus and promotes the transcription of its target genes (Settembre et al, 2011). Aside from lysosomal biogenesis and autophagy, TFEB has been studied in the context of angiogenesis (Doronzo et al, 2019), immune response (Nabar & Kehrl, 2017), epithelial–mesenchymal transition (Huan et al, 2005; Li et al, 2020), and cancer metabolism (Di Malta & Ballabio, 2017), among other processes in various disease-related models. Reprogrammed metabolism is a hallmark of cancer (Hanahan & Weinberg, 2011). Such alterations enable uncontrolled tumor growth or protect against oxidative and metabolic stress (DeBerardinis & Chandel, 2016).

[1]Interdisciplinary Life Sciences Graduate Programs, The University of Texas at Austin, Austin, TX, USA    [2]Livestrong Cancer Institutes, Department of Oncology, Dell Medical School, The University of Texas at Austin, Austin, TX, USA    [3]Division of Chemical Biology and Medicinal Chemistry, College of Pharmacy, The University of Texas at Austin, Austin, TX, USA    [4]Division of Pharmacology and Toxicology, College of Pharmacy, The University of Texas at Austin, Austin, TX, USA

Correspondence: carla.vandenberg@austin.utexas.edu
*Milad Soleimani and Mark Duchow contributed equally to this work

TFEB lies at the intersection of various pathways directly associated with metabolic adaptation. mTORC1 regulates TFEB subcellular localization by phosphorylating S211, S142, and S122 (Vega-Rubin-de-Celis et al, 2017). Amino acid starvation inhibits mTORC1 activity, resulting in the nuclear localization of TFEB. Subsequently, TFEB activates the transcription of lysosomal biogenesis and autophagy genes, which support nutrient catabolism (Martina et al, 2012). Cancers rely on glycolysis for energy production in a process called the Warburg effect. TFEB regulates multiple genes, including *HK1* (Hexokinase 1), *HK2*, *SLC2A1* (Solute Carrier family 2 member 1, a.k.a. GLUT1), and *SLC2A4* (a.k.a. GLUT4), that are involved in glucose metabolism (Mansueto et al, 2017). Glutamine is another source of metabolic support for tumors. It is directly involved in amino acid and nucleotide synthesis as a nitrogen donor and in cellular redox maintenance through glutaminolysis (Son et al, 2013; Kim et al, 2021). TFEB regulates glutamine metabolism by promoting the transcription of glutaminase (Kim et al, 2021). Thus, TFEB may be a nexus for cancer-associated metabolic stress and subsequent cell fate.

In this study, we report a novel role of TFEB in protecting triple-negative breast CSCs against metabolic stress. We have shown that glycolysis inhibitor 2-deoxy-D-glucose (2-DG) inhibits the CSC phenotype. 2-DG–induced stress triggers unfolded protein response (UPR), increasing the expression of UPR-related genes such as BiP/*HSPA5* and CHOP/*DDIT3*. This response is further augmented by nuclear TFEB. In addition, TFEB safeguards CSCs against 2-DG–induced stress by promoting autophagy. Silencing BiP/*HSPA5*, a marker of breast CSCs (Conner et al, 2020), phenocopied TFEB knockdown in terms of mammosphere formation and CD24$^{low}$/CD44$^{high}$ enrichment, supporting that TFEB enriches CSCs by promoting UPR and autophagy.

# Results

## TFEB promotes TNBC self-renewal in vitro

We have previously shown that knocking out TFEB reduces colony formation in TNBC cells (Soleimani et al, 2022). An initial analysis of TFEB expression across various breast cancer subtypes revealed that TNBC displays a higher expression of TFEB than normal mammary tissue and other breast cancer subtypes (Fig 1A) (Chandrashekar et al, 2017). In light of reports citing the ability of CSCs to endure metabolic and oxidative stress (Ciavardelli et al, 2014; Luo et al, 2018) and the function of TFEB in metabolic/oxidative response, we decided to examine the role of TFEB in self-renewal. We performed mammosphere formation and clonogenic assays on shRNA-mediated TFEB knockdown (KD) versus scramble cells. TNBC cell lines, HCC1806, HCC38, MDA-MB-231, and MDA-MB-157, were transduced with either scramble control or TFEB shRNA. Knocking down TFEB significantly inhibited secondary mammosphere formation in TNBC cell lines (Figs 1B and S1A and B). This effect was consistent with the clonogenic assay results that also showed a dramatic decrease in colony formation in TFEB KD cells compared with the control (Figs 1C and S1C). It is widely documented that breast cancer tumors enriched in CD44$^{high}$/

CD24$^{low}$ cells have a high tumor-initiating capacity (DA Cruz Paula & Lopes, 2017). Using data retrieved from Correlation AnalyzeR (Miller & Bishop, 2021), we observed that *TFEB* mRNA expression exhibited a direct correlation to *CD44* levels but no meaningful correlation to *CD24* (Fig 1D). Moreover, knocking down TFEB depleted the CD44$^{high}$/CD24$^{low}$ population in human TNBC cell lines (Fig 1E). The decline in mammosphere and colony formation and the concomitant decrease in CSC markers upon TFEB KD suggest that TFEB plays a role in breast CSC abundance.

Having observed the impact of TFEB KD on cell self-renewal, we decided to assess the transcriptional effects of TFEB in cells. TFEB is phosphorylated by various kinases that regulate its subcellular localization and activity (Puertollano et al, 2018). Phosphorylation of TFEB by mTORC1 at S142 and/or S211 sequesters it in the cytoplasm, whereas the absence of phosphorylation at these sites results in the nuclear translocation of TFEB (Puertollano et al, 2018). As a transcription factor, nuclear TFEB induces the transcription of its target genes. To perform a gain-of-function study of active TFEB, we used a mutant TFEB construct harboring an alanine in place of serine at position 142 (Settembre et al, 2011). First, TFEB (S142A) was overexpressed in cells and validated by Western blot to have a predominantly nuclear localization (Fig 1F). In addition, the ectopic expression of TFEB (S142A) increased mammosphere formation compared with control in TNBC cells (Fig 1G).

## TFEB knockdown suppresses TNBC self-renewal in vivo

To assess the impact of TFEB KD on tumor growth kinetics in vivo, mice were orthotopically injected with either scramble control or TFEB KD HCC1806 cells. Over 20 d, TFEB KD cells displayed significantly slower growth than their scramble counterparts (Fig 2A). We carried out a limiting dilution assay, the gold standard of CSC evaluation, to determine whether TFEB KD affects tumor-initiating capacity in vivo. Mice were injected with either scramble control or TFEB KD HCC1806 cells serially diluted at $5 \times 10^5$, $5 \times 10^4$, $5 \times 10^3$, or $5 \times 10^2$. Tumor growth was monitored for 140 d after injection. ELDA (extreme limiting dilution analysis) analysis demonstrated that TFEB KD cells had an ~10-fold lower tumor-initiating cell frequency compared with control cells (Fig 2B). Hematoxylin and eosin staining revealed a pattern of cellularity in TFEB KD tumors that was less dense than that of control tumors (Fig 2C).

## 2-DG inhibits TNBC growth and self-renewal

Normal cells depend primarily on oxidative phosphorylation for their metabolic needs, whereas cancer cells resort to aerobic glycolysis in a phenomenon called the Warburg effect (Liberti & Locasale, 2016). Evidence indicates that the transition from oxidative phosphorylation to glycolysis promotes cancer stemness in breast cancer (Dong et al, 2013). Others have demonstrated that blocking glycolysis with 2-DG reduces breast CSC populations (Luo et al, 2018). The mechanism whereby 2-DG impacts CSC character has yet to be determined. To assess the effect of 2-DG on cell viability, TNBC cell lines HCC1806, MDA-MB-231, MDA-MB-157, MDA-MB-453, HCC1937, HCC38, BT549, SW527, and HCC70 were treated with either vehicle or 0.3–20 mmol/l 2-DG for 72 h and analyzed using the CellTiter-Glo cell viability assay. All cell lines displayed a dose-

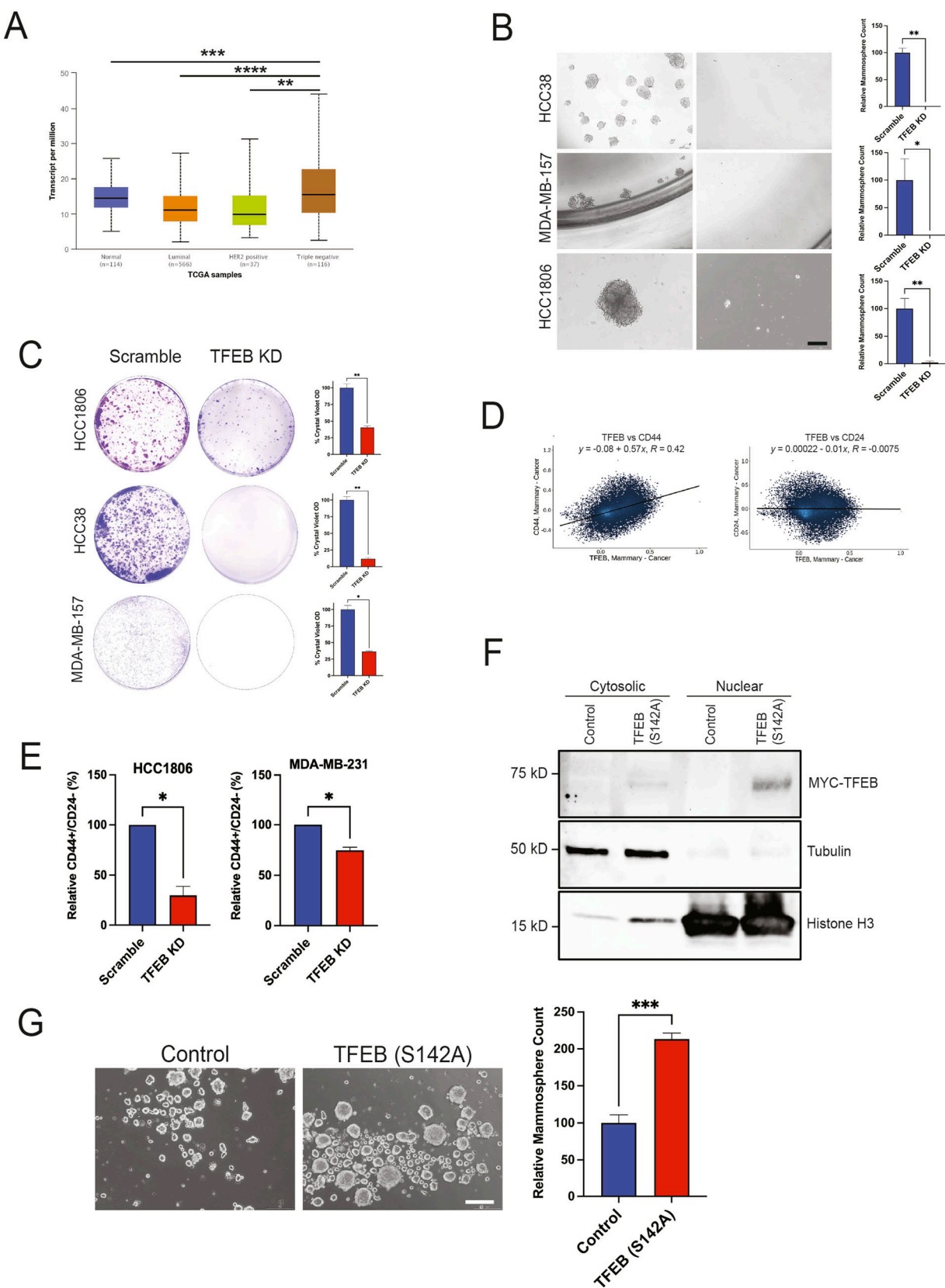

dependent decline in cell viability in response to 2-DG (Fig 3A). That being said, some cell lines, such as MDA-MB-157 and HCC70, were less sensitive than others, such as HCC1806 and HCC1937 (Fig 3A). Similar trends were observed in colony formation where cells were treated with either vehicle or 1, 2, 5, 10, or 15 mmol/l 2-DG for 72 h and recovered for 7–10 d (Figs 3B and S2A). To determine how 2-DG treatment impacts self-renewal in vitro, TNBC cell lines HCC1806, HCC38, and MDA-MB-157 were incubated in mammosphere media and treated with either vehicle or 2-DG. Each cell line was treated with its corresponding ~ IC-50 dose of 2-DG derived from data in Fig 3A. As expected, 2-DG suppressed self-renewal in every cell line tested, as illustrated by the significantly reduced mammosphere growth (Fig 3C).

To further verify the inhibition of CSC character by 2-DG, we examined various CSC biomarkers. We started by analyzing CD44$^{high}$/CD24$^{low}$ cells in HCC1806, HCC38, and HCC1937. Cells were incubated either in the absence or in the presence of 1, 2, 5 mmol/l 2-DG for 24 h. There was a dose-dependent decline in CD44$^{high}$/CD24$^{low}$ cells (Figs 3D and S2B). Next, we interrogated another CSC biomarker CD49f (a.k.a. integrin alpha-6 [ITGA6]) in response to 2-DG treatment. Others have documented a close association between self-renewal and CD49f expression in human breast cancer (To et al, 2010). Our observations pointed to a striking reduction in CD49f levels, similar to those of CD44$^{high}$/CD24$^{low}$, in HCC1806 and HCC38 cells exposed to 2-DG for 24 h (Fig 3E). Together, we have shown, both phenotypically and using CSC biomarkers, that 2-DG–induced metabolic stress diminishes TNBC stem cell populations.

## 2-DG inhibits TFEB phosphorylation and induces TFEB nuclear translocation

We highlighted earlier that TFEB promotes cancer stemness in TNBC (Figs 1 and 2). TFEB plays an instrumental role in responding to metabolic stress (Martina et al, 2016). These observations led us to investigate whether TFEB regulates cellular response to glycolysis inhibitor 2-DG in TNBC. Preliminary Western blot analysis of TNBC cells treated with increasing concentrations of 2-DG revealed a gradual decrease in TFEB levels and a downward electrophoretic shift in TFEB bands (Fig S3A). Loss of phosphorylation is accompanied by proteasomal degradation (Sha et al, 2017). The shift in TFEB mobility is indicative of reduced phosphorylation. To identify the phosphorylation sites impacted by glycolytic stress, we performed a time course where TNBC cells were glucose-starved for 3, 8, or 24 h. Western blot analysis revealed that glucose starvation inhibited TFEB phosphorylation at positions S211 and S122 (Fig S3A). Unlike TFEB, TFE3 did not show a consistent response to glucose starvation (Fig S3A). We have previously shown that inhibiting TFEB phosphorylation results in its nuclear translocation (Soleimani

et al, 2022). Indeed, subcellular fractionation of cells treated with either vehicle or 2-DG revealed TFEB nuclear localization in the presence of 2-DG (Fig S3B). The dephosphorylation and nuclear translocation of TFEB normally indicate its activation of downstream transcriptional targets (Puertollano et al, 2018). Others have shown that the CLEAR (coordinated lysosomal expression and regulation) motif is a reliable regulatory target to assess TFEB activity (Cortes et al, 2014). To verify increased TFEB activity in response to 2-DG, we performed a promoter/reporter assay using the 4xCLEAR-luciferase reporter and pRL SV40 *Renilla* luciferase constructs in cells subjected to 2-DG treatment, glucose starvation, or no treatment (Fig 3F). In concordance with the dephosphorylation and localization results, the reporter assay pointed to an increase in TFEB activity in response to 2-DG treatment (Fig 3F). Glucose starvation had a similar effect on TFEB activity, reaffirming the glycolytic mechanism of 2-DG response (Fig 3F). TFEB activity and localization are regulated by multiple kinases, including mTOR, ERK, and AKT (Puertollano et al, 2018). To evaluate the effect of 2-DG on mTOR signaling in TNBC, we looked at mTORC1 target 4E-BP1 and mTORC2 targets AKT and Rictor. Surprisingly, 2-DG did not have a consistent impact across all cell lines on mTOR signaling at the concentrations used (Figs 3G and S3C). Another kinase upstream of TFEB is 5′-AMP–activated protein kinase (AMPK) (El-Houjeiri et al, 2019). AMPK activity increases in response to reduced ATP:AMP ratios induced by glycolytic stress. Unlike mTOR, AMPK showed a robust response to 2-DG treatment through phosphorylation of its well-known target acetyl-CoA carboxylase (ACC) (Figs 3F and S3C). These data demonstrate that 2-DG–driven stress increases TFEB transcriptional activity and AMPK activity.

## 2-DG induces UPR

UPR is a stress mechanism to reduce the accumulation of unfolded or misfolded proteins causing ER stress. 2-DG is known to cause ER stress (Yu & Kim, 2010). UPR occurs via three major ER membrane stress sensors PERK (protein kinase R-like ER kinase; a.k.a. EIF2AK), IRE1α (inositol-requiring transmembrane kinase/endoribonuclease 1α), and Activating Transcription Factor 6 (ATF6) (Hetz et al, 2020). Various components of UPR are closely associated with breast CSC character, suggesting that UPR preserves CSC populations. Metastatic breast tumors, with high levels of CD44$^{high}$/CD24$^{low}$ cells, display an up-regulation of binding immunoglobulin protein (BiP; a.k.a. GRP78, HSPA5) and protein disulfide isomerase (PDI) (Bartkowiak et al, 2010). Inhibition of ATF6 and PERK suppresses mammosphere formation (Li et al, 2018). Knockdown of X-box Binding Protein 1 (*XBP1*) reduces CD44$^{high}$/CD24$^{low}$ enrichment in TNBC (Chen et al, 2014). The overexpression of BiP increases CD44$^{high}$/CD24$^{low}$ cells and up-regulates CSC-

**Figure 1.   TFEB enriches TNBC CSC populations in vitro.**
**(A)** UALCAN analysis of TFEB mRNA expression across breast cancer subtypes (one-way ANOVA, Dunnett's test). Scale bar: 200 $\mu$m. **(B)** Mammosphere formation assay of indicated TNBC cell lines transduced with either scramble or TFEB shRNA and grown in mammosphere media for 7–10 d, followed by passaging to form secondary mammospheres. **(C)** Clonogenic assay of indicated cell lines transduced with either scramble or TFEB shRNA. **(D)** Scatter plots of TFEB versus CD44 and TFEB versus CD24 mRNA expression in breast cancer tumors as retrieved by Correlation AnalyzeR. **(E)** Flow cytometric analysis of the CD44$^{high}$/CD24$^{low}$ fraction in indicated TNBC cell lines transduced with either scramble or TFEB shRNA (*t* test). **(F)** Western blot analysis of cytosolic/nuclear fractions in HCC1806 expressing either empty pLenti-TrueORF or TFEB (S142A). **(G)** Representative images and quantification of secondary HCC1806 mammospheres expressing either empty control or TFEB (S142A) (*t* test). Scale bar: 200 $\mu$m. *$P$ < 0.05; **$P$ < 0.01; ***$P$ < 0.001; ****$P$ < 0.0001.

associated genes (Conner et al, 2020). PERK knockdown in mouse mammary carcinoma cells reduces tumor initiation and expansion (Bobrovnikova-Marjon et al, 2010). In this study, we measured the change in the expression of several UPR markers caused by 2-DG exposure, namely, *XBP1*, DNA damage–inducible transcript 3 (*DDIT3*, a.k.a. CHOP), Activating Transcription Factor 4, PDI family A member 2 (*PDIA2* a.k.a. PDI), ER degradation–enhancing α-mannosidase-like protein 1 (*EDEM1*), protein phosphatase 1 regulatory subunit 15A (*PPP1R15A*), heat shock 70 kD protein 5 (*HSPA5*), and *ATF6* in HCC1806 and HCC38 using qRT–PCR (Fig 4A). We further confirmed UPR activation by 2-DG using Western blot. To that end, HCC1806, HCC70, BT549, MDA-MB-157, and HCC1937 cells were treated with either vehicle or increasing concentrations of 2-DG. The results revealed UPR induction as shown by an up-regulation of PERK, PDI, BiP, CHOP, and IRE-1α (Fig 4B). Tunicamycin (TM), a well-documented ER stress inducer, mostly mirrored the effect of 2-DG on UPR (Fig 4C).

### TFEB mediates 2-DG–driven UPR

First, we investigated the role of TFEB in responding to 2-DG–induced metabolic stress and the resulting impact on CSCs. Normally, TFEB resides in the cytoplasm bound to the lysosomal membrane through interaction with a component of the Ragulator complex but localizes to the nucleus under various types of stress. To determine the significance of TFEB subcellular localization, we generated cell lines expressing either predominantly nuclear or cytosolic TFEB: (1) TFEB (S142A) is a constitutively nuclear TFEB mutant (Fig 1F), and (2) RagC (S75L) sequesters TFEB in the cytoplasm (Fig 5A). First, we performed a mammosphere formation assay with HCC1806 cells transduced with TFEB (S142A), RagC (S75L), or empty vector treated with either vehicle or 2-DG and incubated for 7–10 d. The stable overexpression of TFEB (S142A) lowered 2-DG sensitivity in TNBC mammospheres compared with control. In contrast, the overexpression of RagC (S75L) enhanced 2-DG cytotoxicity (Fig 5B). Overexpressing TFEB (S142A) in two other cell lines, HCC1937 and HCC38, mirrored its effect in HCC1806 in terms of lowering 2-DG sensitivity (Fig S3D). Overall, these findings align with the observation that nuclear TFEB positively regulates CSC self-renewal in TNBC.

Given that both TFEB and UPR enhance cancer stemness (Spaan et al, 2019; Liang et al, 2021) and are responsive to metabolic stress, we hypothesized that TFEB modulates 2-DG–induced UPR. To test this hypothesis, we compared UPR activation by 2-DG in scramble control and TFEB KD cells. A Western blot analysis showed that silencing TFEB decreased 2-DG up-regulation of UPR markers CHOP, BiP, PERK, and IRE1α (Fig 5C). Indeed, the overexpression of constitutively nuclear TFEB (S142A) augmented up-regulation of BiP and CHOP at both protein (Fig 5D) and mRNA (Fig 5E) levels, whereas RagC (S75L) slightly lowered 2-DG–driven BiP up-regulation (Fig 5D). Interestingly, TFEB KD diminished UPR induction, and TFEB (S142A) enhanced it in cells treated with tunicamycin (Fig 5C and D). As clinical correlates, *TFEB* mRNA expression in TNBC patient samples showed a direct correlation of TFEB expression to *EIF2AK*, *DDIT3*, and *HSPA5* (Fig S3E).

### BiP/*HSPA5* knockdown suppresses the CSC phenotype

We have shown that 2-DG activates a TFEB-UPR axis while suppressing cancer stemness. Others have reported that BiP/*HSPA5* overexpression

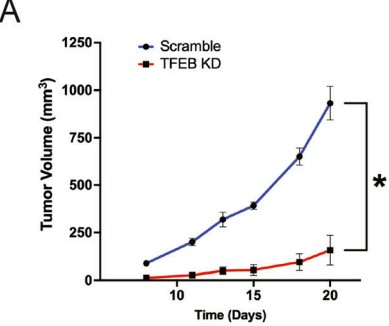

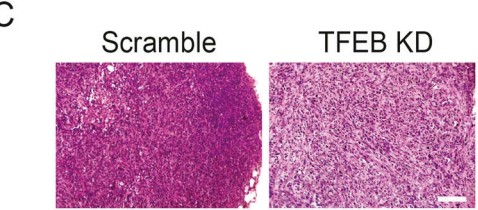

**Figure 2. TFEB knockdown reduces TNBC tumor growth and self-renewal in vivo.**
**(A)** Average tumor growth curves of HCC1806 cells transduced with either scramble (n = 10) or TFEB shRNA (n = 10) and injected orthotopically into nude mice (*t* test, *$P < 0.05$). **(B)** Tumor-limiting dilution assay of HCC1806 cells transduced with scramble or TFEB shRNA. NSG mice were orthotopically injected with $5 \times 10^5$ (n = 5), $5 \times 10^4$ (n = 6), $5 \times 10^3$ (n = 7), or $5 \times 10^2$ (n = 9) cells. **(C)** Representative H&E images of either scramble control or TFEB KD HCC1806 xenograft tumors. Scale bar: 100 μm.

(a key UPR-associated gene) increases CD44[high]/CD24[low] cells in breast cancer (Conner et al, 2020). We aimed to determine whether silencing BiP/*HSPA5*, also a TFEB-responsive gene, impacts TNBC self-renewal. To validate the shRNA-mediated BiP/*HSPA5* KD, we performed a Western blot on scramble and BiP/*HSPA5* KD cells treated with either vehicle or 2-DG for 24 h. BiP/*HSPA5* KD attenuated BiP/*HSPA5* and CHOP up-regulation in response to 2-DG (Fig 6A). In untreated TNBC cells, BiP/*HSPA5* KD inhibited self-renewal as indicated by significantly reduced colony and mammosphere formation (Fig 6B and C). Furthermore, an analysis of CSC biomarkers revealed a decline in CD44[high]/CD24[low] cells upon BiP/*HSPA5* KD, consistent with a diminished CSC population (Fig 6D).

### TFEB and UPR promote autophagy in response to 2-DG

Thus far, our data show that 2-DG induces UPR and TFEB activity. However, we lacked an explanation as to how TFEB rescues CSCs

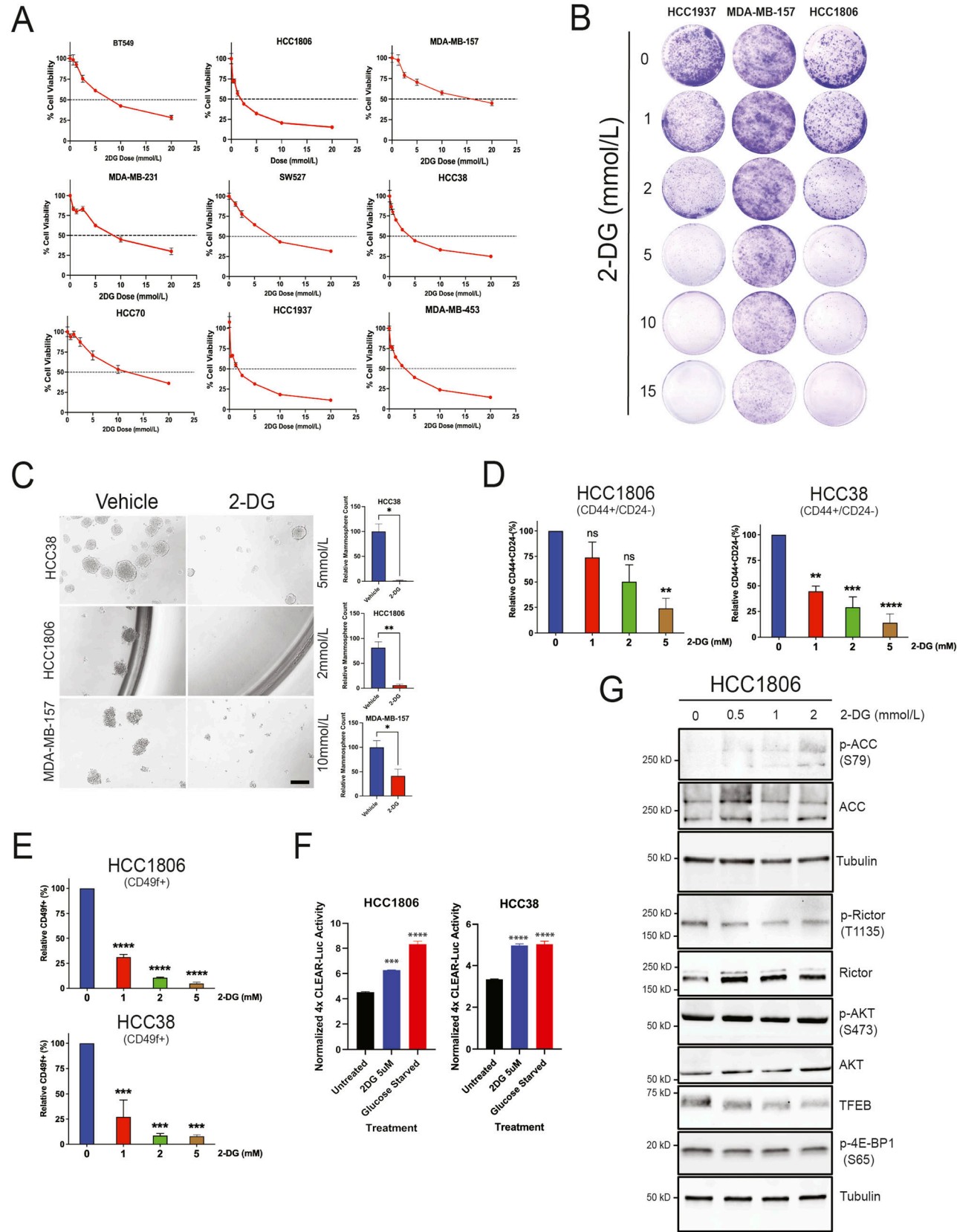

from 2-DG–induced stress. CSCs and tumor cells use UPR as a survival response and can engage autophagy (ER-phagy and mitophagy) to clear unfolded proteins and damaged organelles (Senft & Ronai, 2015). We anticipated TFEB might shift cells toward an autophagic survival response because TFEB directly regulates the expression of autophagy- and lysosome-related genes (Settembre et al, 2011). We tested whether 2-DG induces autophagy, as shown by increased p62 and LC3-II compared with vehicle controls. Indeed, there was an increase in p62 and LC3-II levels upon 2-DG treatment (Fig 7A). TFEB KD reduced the autophagic response to 2DG (Fig 7A). In addition, cotreatment with the PERK inhibitor ISRIB to reduce UPR showed a similar response to TFEB KD, attenuating the autophagic response to 2-DG (Fig 7B). We validated the effect of ISRIB on autophagy by assessing LC3 using ICC (Fig 7C). Hydroxychloroquine (HXQ) was used as a positive control to cause LC3 puncta. Together, these data support that 2-DG and TFEB increase the expression of several UPR genes, whereas TFEB promotes autophagy to sustain cancer and CSC populations.

## Discussion

The unique metabolic landscape of CSCs represents a therapeutic opportunity. Cancer cells undergo metabolic alterations to meet their energetic needs for growth. These alterations result in a glycolytic mode of metabolism accompanied by an increase in glucose uptake (Hanahan & Weinberg, 2011). CSCs, on the contrary, are metabolically flexible in response to the tumor microenvironment (Snyder et al, 2018). One hypothesis is that CSCs follow the hierarchical pattern of normal cells where differentiation is closely associated with a switch from glycolysis to oxidative phosphorylation (Sancho et al, 2016). Others have postulated that CSC character is driven by metabolic reprogramming in cells (Menendez & Alarcon, 2014). A comprehensive model of non-CSC-to-CSC transition showed that intermediary metabolites, such as $\alpha$-ketoglutarate and acetyl-CoA, govern the epigenetic regulation of genes involved in CSC character (Menendez & Alarcon, 2014). In breast cancer, researchers have reported the significance of glycolytic shift and enhanced macromolecule biosynthesis to cancer stemness maintenance (Dong et al, 2013). Several seemingly contradictory reports point to either glycolysis (Zhou et al, 2011; Ciavardelli et al, 2014) or oxidative phosphorylation (Janiszewska et al, 2012) as the preferred mode of CSC metabolism. Understanding how metabolic stress modulates CSC dynamics is key to developing an effective treatment that targets the entire CSC population.

Given the central role that autophagy and lysosomes play in cellular metabolism and that TFEB is a master regulator of genes involved in metabolic adaptation, we decided to explore whether TFEB controls CSC populations with or without metabolic stress. Herein, the knockdown of TFEB in TNBC cells strongly inhibited mammosphere formation and tumor initiation. First, we used CSC biomarkers CD44$^{high}$/CD24$^{low}$ and in vitro and in vivo functional assays to assess the role of TFEB in CSCs regardless of metabolic stress. We also found TFEB pivotal in safeguarding CSCs against metabolic stress. Our study specifically indicates that TFEB attenuates the suppressive effects of 2-DG–induced stress on TNBC self-renewal by promoting autophagy and UPR. Autophagy involves, among other things, the lysosomal metabolism of sulfur amino acids, which contributes to the cellular cysteine reservoir for antioxidant defense and adaptation (Matye et al, 2022). mTOR-driven aberrant suppression of autophagy sensitizes cisplatin-resistant lung cancer to 2-DG (Gremke et al, 2020). Another instance of TFEB responding to metabolic stress is its induction of mitophagy to counteract metabolic/mitochondrial stress-driven Ca$^{2+}$ release to support pancreatic $\beta$-cell function (Park et al, 2022). Altogether, TFEB and the associated metabolic stress response machinery represent a potential vulnerability in CSCs and a promising area for therapeutic exploration. It is also important to point out that either the RNA or protein abundance of TFEB is less likely to serve as a biomarker of its activity than its subcellular localization. Our studies and others bear out this conclusion (Zhu et al, 2021).

We have demonstrated that 2-DG suppresses mammosphere formation and CD44$^{high}$/CD24$^{low}$ cells associated with mesenchymal-like CSCs in TNBC. It bears mentioning that chronic metabolic stress promotes breast cancer stemness in a Wnt-dependent fashion (Lee et al, 2015) and that 2-DG promotes ALDH + epithelial-like CSCs while inhibiting invasive mesenchymal-like CSCs (Luo et al, 2018). Epithelial-like CSCs are usually proliferative, whereas mesenchymal-like CSCs are dormant (Luo et al, 2018). It is important to consider how each group contributes to tumor initiation versus growth when studying CSCs. Hyperactivation of UPR and/or autophagy as a consequence of 2-DG treatment could lead to increased CSCs. Furthermore, it is critical to acknowledge that each CSC assay alone is insufficient to accurately assess CSC character. Therefore, we have used a combination of CD44$^{high}$/CD24$^{low}$ and CD49f+ cells as biomarkers. In addition, we have used functional assays in the form of clonogenic and mammosphere assays in vitro and a tumor-limiting dilution assay in vivo. Using several different methods is vital to ensure the reliability and robustness of CSC data.

A mechanistic analysis of our results revealed that TFEB-directed 2-DG response occurs via UPR. In breast cancer, UPR is documented to stimulate cancer stemness (Liang et al, 2021). 2-DG induced UPR in TNBC cells in a dose-dependent fashion. This effect was mitigated upon TFEB KD and amplified upon TFEB (S142A) overexpression. We have shown that BiP/*HSPA5* and CHOP/*DDIT3* are the most consistently up-regulated UPR markers at both mRNA and

**Figure 3.    2-DG suppresses cell viability and cancer stem cell phenotype in TNBC.**
**(A)** Cell viability assay of indicated TNBC cell lines treated with either vehicle or 2-DG (0.3–20 mmol/l) for 72 h. **(B)** Clonogenic assay of indicated TNBC cell lines treated with either vehicle or 2-DG (1, 2, 5, 10, and 15 mmol/l) for 72 h and recovered for 7–10 d. **(C)** Mammosphere formation assay of indicated TNBC cell lines treated with either vehicle or 2-DG for 7–10 d. Scale bar: 200 $\mu$m. **(D, E)** Flow cytometric analysis of CD44$^{high}$/CD24$^{low}$ and (E) CD49f fractions in indicated TNBC cell lines treated with either vehicle or 2-DG for 24 h (one-way ANOVA, Dunnett's test). **(F)** CLEAR reporter assay of HCC1806 and HCC38 2-DG–treated (5 mmol/l), glucose-starved, or untreated cells for 24 h (one-way ANOVA, Dunnett's test). **(G)** Western blot analysis of TFEB, p-4E-BP1 (S65), p-Rictor (T1135), Rictor, p-AKT (S473), AKT, p-ACC (S79), and ACC in HCC1806 treated with either vehicle or 2-DG for 24 h. $*P < 0.05$; $**P < 0.01$; $***P < 0.001$; $****P < 0.0001$.

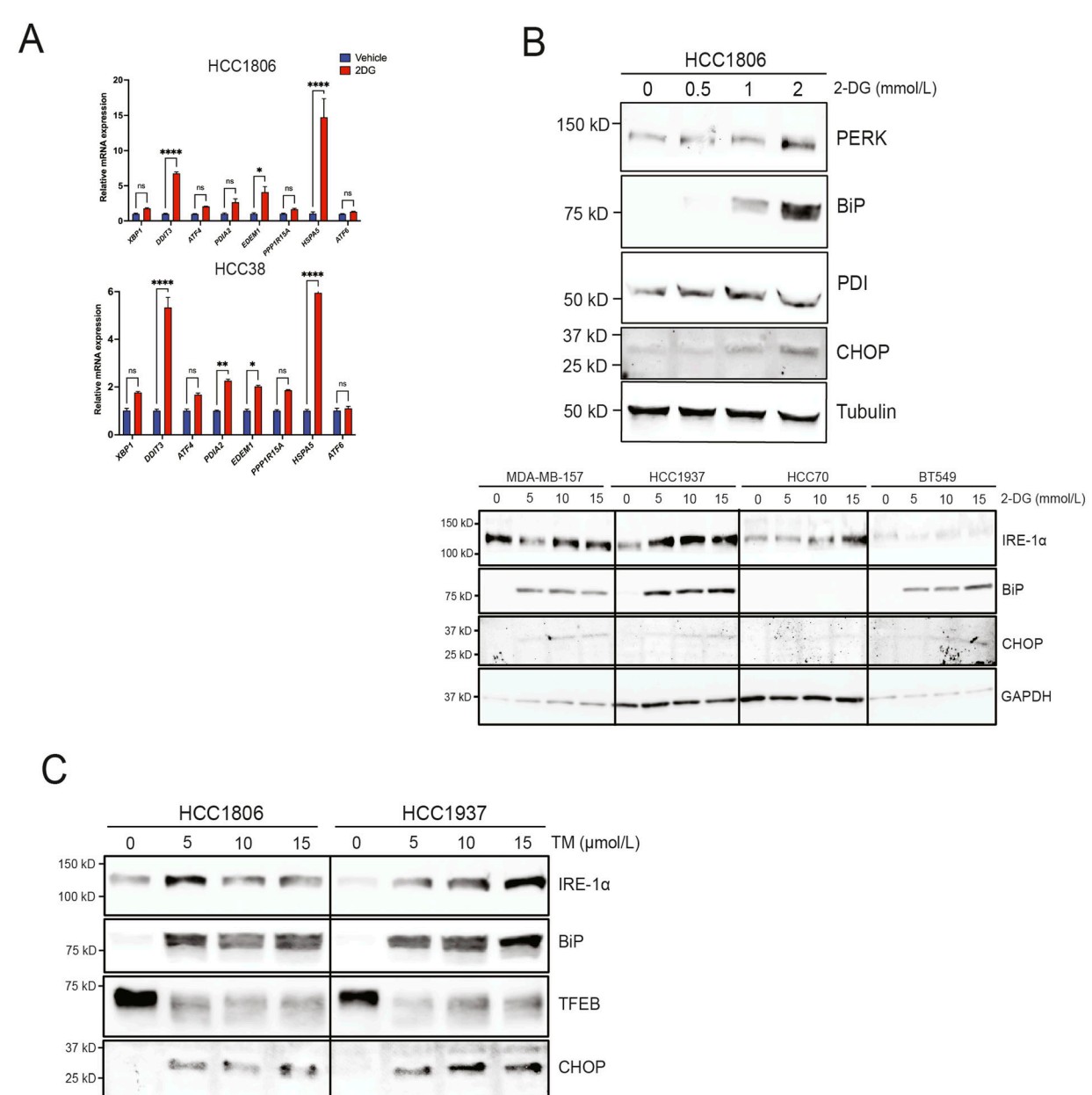

**Figure 4. 2-DG induces unfolded protein response (UPR).**
**(A)** qRT–PCR analysis of indicated UPR markers in HCC1806 and HCC38 treated with either vehicle or 2-DG for 24 h (two-way ANOVA, Sidak's test). **(B)** Western blot analysis of UPR markers in indicated TNBC cell lines treated with either vehicle or increasing concentrations of 2-DG for 24 h. **(C)** Western blot analysis of UPR markers in indicated TNBC cell lines treated with either vehicle or increasing concentrations of tunicamycin. Veh, vehicle; TM, tunicamycin. *$P < 0.05$; **$P < 0.01$; ***$P < 0.001$; ****$P < 0.0001$.

protein levels. Furthermore, these were the two most altered markers because of either TFEB KD or overexpression. There are several UPR markers implicated in CSC regulation. BiP localized to the cell surface promotes CSC phenotype and metastasis in breast cancer (Conner et al, 2020). It is up-regulated in bone marrow–derived disseminated breast tumor cells displaying CSC character (Bartkowiak et al, 2010). A meta-analysis of BiP and its clinicopathological potential in breast cancer found a correlation between high BiP expression and HER2 and basal-like

subtypes, as well as metastatic tumors (Direito et al, 2022). XBP1 is critical for the tumorigenicity, progression, and relapse of TNBC, where it forms a complex with hypoxia-inducible factor 1 subunit $\alpha$ to maintain CSCs (Chen et al, 2014). TNBC/basal-like tumors have higher levels of TFEB than the other breast cancer subtypes. This renders studying TFEB and its regulation of metabolic stress and CSCs in TNBC more clinically relevant. Although TNBC patients generally respond to chemotherapy, they have an earlier relapse and a more frequent recurrence

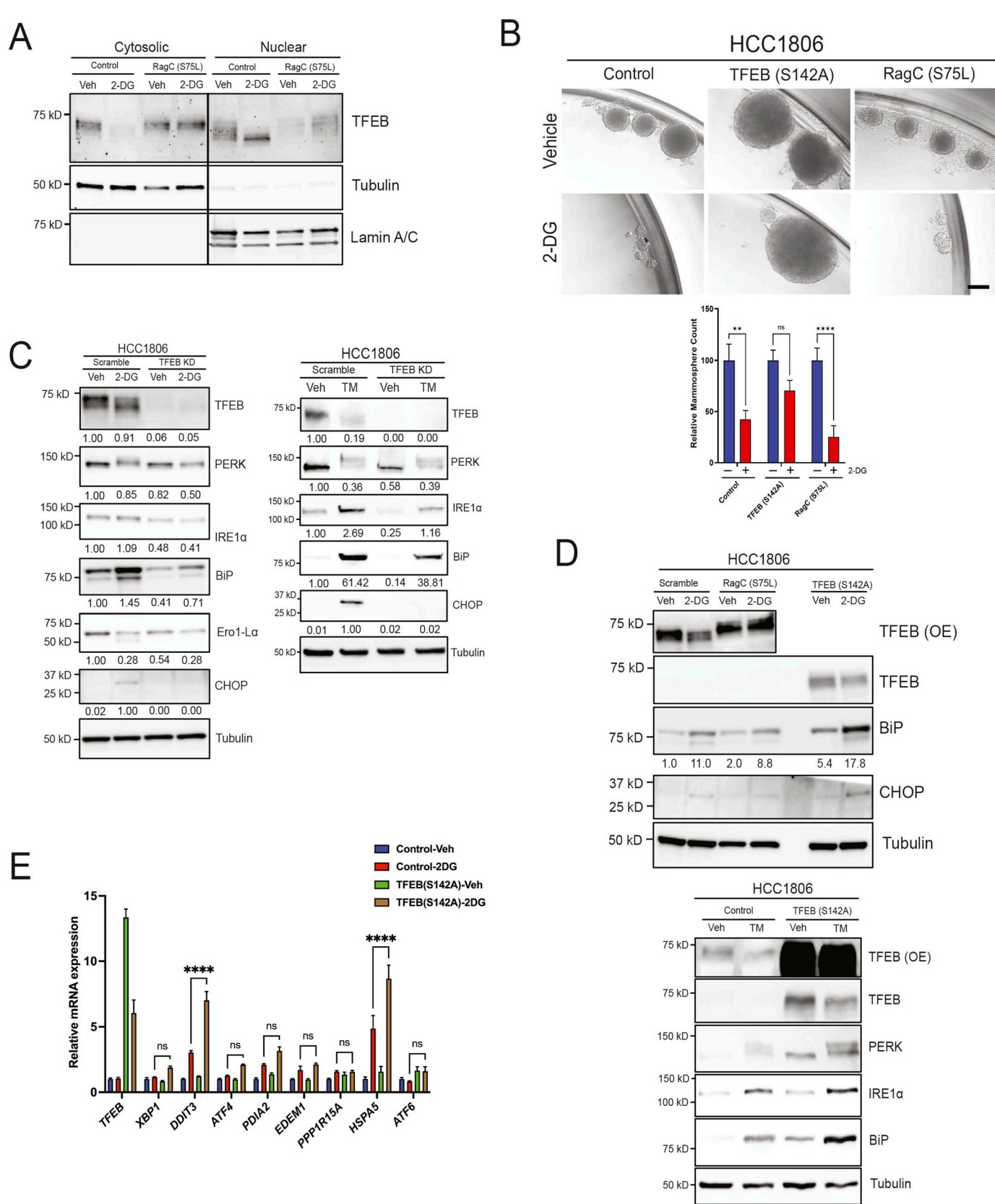

than the other breast cancer subtypes (Zagami & Carey, 2022). This is partly attributable to the dormant CSC population that is insensitive to most therapies but gives rise to tumor heterogeneity. Thus, new therapeutics must target promiscuous CSCs to overcome the barriers associated with tumor heterogeneity, treatment resistance, metastasis, and tumor recurrence.

In conclusion, we have uncovered a novel metabolic stress response mechanism where TFEB sustains CSCs by up-regulating UPR and autophagy in TNBC. TFEB depletion suppressed self-renewal in vitro and in vivo, and the overexpression of active TFEB thwarted 2-DG–induced mammosphere inhibition. Mechanistically, TFEB KD cells showed a diminished UPR response to 2-DG, whereas TFEB (S142A)-overexpressing cells had a more robust response than the corresponding controls. The key role of TFEB during metabolic stress appears to be as a UPR-responsive gene that promotes cell survival by enhancing autophagy and other key UPR-related genes that regulate TNBC CSCs, namely, BiP. Our limited clinical analyses further support a metabolic gene signature involving TFEB regulation of CSC and UPR markers. This further highlights the importance of pursuing these pathways to better understand CSC biology and potential new targets for treating TNBC.

# Materials and Methods

### Cell culture and reagents

TNBC cell lines MDA-MB-231, MDA-MB-157, MDA-MB-453, HCC1806, HCC70, HCC38, SW527, HCC1395, HCC1937, and BT549 were obtained from the American Type Culture Collection (ATCC). The cell lines from the ATCC were grown in either RPMI 1640 (Gibco) or DMEM (Gibco) with FBS, GlutaMAX (Gibco), and penicillin/streptomycin at 37°C, 5% $CO_2$. Tunicamycin (T7765) and 2-DG (25972) were purchased from Millipore-Sigma. ISRIB (S7400) was purchased from Selleck Chemicals.

### Western blot

Antibodies against TFEB (4240), p-TFEB (S122) (87932), p-TFEB (S211) (37681), 4E-BP1 (9644), p-4E-BP1 (S65) (9451), p-4E-BP1 (T37/46) (2855), p-Rictor (2114), p-Rictor (T1135) (3806), p-ACC (3661), ACC (3676), AKT (2920), p-AKT (S473) (4060), LC3 (12741), p62 (SQSTM1) (88588), Ero1-Lα (3264), BiP (3177), IRE1α (3294), PDI (3501), CHOP (2895), PERK (5683), MYC (2276), GAPDH (5174), histone H3 (14269), and lamin A/C (4777) were purchased from Cell Signaling Technology. Total cell lysates were prepared using ice-cold RIPA buffer

(Thermo Fisher Scientific) supplemented with Halt Protease and Phosphatase Inhibitor Cocktail (Thermo Fisher Scientific). Protein concentration was quantified using DC Protein Assay (Bio-Rad). Equal amounts of lysates were loaded onto the SDS–polyacrylamide gel and subsequently transferred to a PVDF/0.2-µm nitrocellulose membrane (Bio-Rad). Blots were incubated overnight with primary antibodies diluted at 1:1,000 in 5% non-fat milk at 4°C. Secondary antibodies anti-rabbit IgG-HRP (7074; Cell Signaling Technology), StarBright Blue 700 anti-rabbit IgG (12004162; Bio-Rad), StarBright Blue 700 anti-mouse IgG (12004159; Bio-Rad), or DyLight 488 anti-mouse IgG (STAR117D488 GA; Bio-Rad) were diluted in 5% non-fat milk. Either GAPDH (5174; Cell Signaling Technology) or hFAB rhodamine anti-tubulin (12004165; Bio-Rad) was used as a loading control.

### Plasmids and transfection

The following plasmids were obtained from Addgene: psPAX2 (12260; Dr. Didier Trono), pCMV-VSV-G (8454; Dr. Bob Weinberg), pRK5-HA GST RagC 75L (19305; Dr. David Sabatini), and 4XCLEAR-luciferase reporter (66800; Dr. Albert La Spada). TFEB (TRCN0000013109; TRCN0000013108) and HSPA5 (TRCN0000001024) shRNAs were purchased from Millipore-Sigma. TFEB (S142A) was a gift from Dr. Andrea Ballabio at Telethon Institute of Genetics and Medicine, Italy. TFEB (S142A) and Rag C (S75L) were cloned into pLenti-C-Myc-DDK-IRES-Puro Lentiviral Gene Expression Vector (OriGene). HEK293T cells were transfected using Lipofectamine 3000 (Invitrogen), and lentiviral particles were harvested to transduce target cell lines.

### Microscopy and immunocytochemistry

Brightfield (BF) imaging was done on an Olympus CKX41 inverted microscope. Paraffin-embedded samples were sectioned, stained with hematoxylin and eosin, and imaged on a Nikon Eclipse Ni-E upright microscope.

Immunocytochemistry (ICC) was performed as previously described (Soleimani et al, 2022). Primary antibody LC3 (12741; Cell Signaling Technology) was used in 5% goat serum at 4°C overnight. The secondary antibody Alexa Fluor 555 anti-rabbit IgG (4413; Cell Signaling Technology) was used in 5% goat serum for 2 h at room temperature. Next, cells were counter-stained with DAPI and imaged using a Nikon Eclipse Ti2 A1R confocal microscope.

### Promoter reporter assay

Cells were cotransfected with 4xCLEAR-luciferase reporter construct and pRL SV40 *Renilla* luciferase construct using

---

**Figure 5. TFEB mediates 2-DG–driven UPR.**
**(A)** Western blot analysis of nuclear/cytosolic fractions of either pLenti or RagC (S75L) treated with either vehicle or 2-DG for 24 h. Tubulin was used as a cytosolic marker and lamin A/C as a nuclear marker. **(B)** Mammosphere formation assay of empty pLenti, TFEB (S142A), or RagC (S75L) HCC1806 treated with either vehicle or 2-DG (two-way ANOVA, Sidak's test). Mammospheres ≥50 µm were counted and normalized to the vehicle control of each group. Scale bar: 200 µm. **(C, D)** Western blot analysis of UPR markers in either scramble or TFEB KD, and (D) empty pLenti, RagC (S75L), or TFEB (S142A) treated with either vehicle, 2-DG, or tunicamycin for 24 h. **(E)** qRT–PCR analysis of UPR markers in either empty pLenti or TFEB (S142A) cells treated with either vehicle or 2-DG (2 mmol/l) for 24 h (two-way ANOVA, Sidak's test). Veh, vehicle; TM, tunicamycin. *$P < 0.05$; **$P < 0.01$; ***$P < 0.001$; ****$P < 0.0001$.

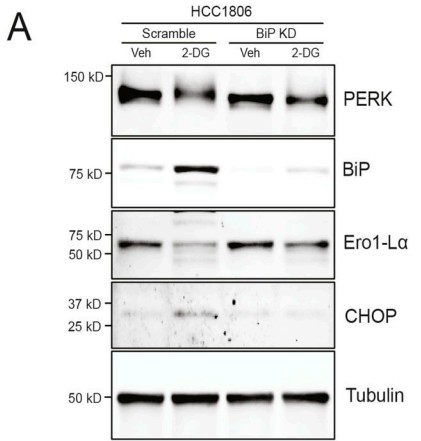

**A**

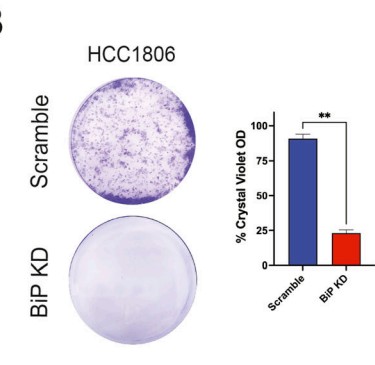

**B**

**C**

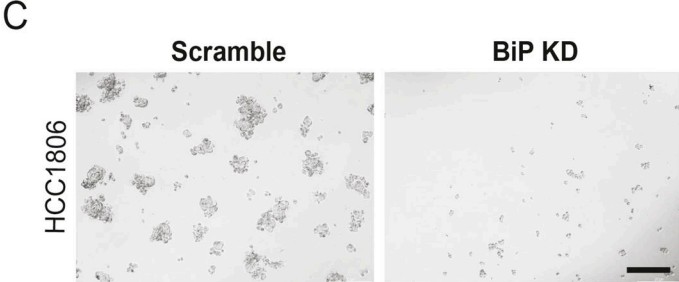

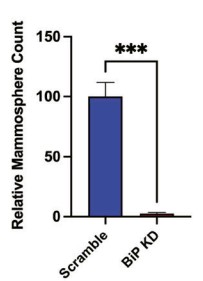

**D**

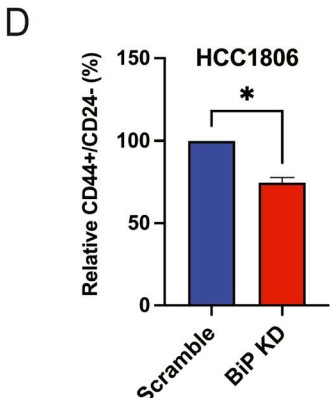

**Figure 6. BiP/HSPA5 knockdown suppresses TNBC self-renewal.**
**(A)** Western blot of indicated UPR markers in either scramble or BiP/HSPA5 KD cells treated with either vehicle or 2-DG for 24 h. **(B, C)** Clonogenic assay and (C) mammosphere formation assay of indicated cell lines transduced with either scramble or BiP/HSPA5 shRNA. Scale bar: 200 μm. **(D)** Flow cytometric analysis of CD44high/CD24low cells in HCC1806 transduced with either scramble or TFEB shRNA (t test, *P < 0.05). Veh, vehicle.

Lipofectamine 3000 (Invitrogen) for 24 h. After transfection, cells were incubated for 24 h in one of three conditions: untreated, 5 mmol/l 2-DG, or glucose-free culture media. After treatment, cells were lysed and luciferase activity was quantified in a plate reader (BioTek).

### Flow cytometry

Cells were trypsinized and washed with the flow wash buffer comprising PBS supplemented with FBS and EDTA. Primary antibodies against CD44 (338806; BioLegend), CD24 (311104; BioLegend), and CD49f (313612; BioLegend) were diluted in the flow wash buffer and incubated with cells for 30 min at room temperature. After washing with the flow wash buffer, cells were resuspended in propidium iodide and analyzed by flow cytometry (Cytek Aurora).

### qRT–PCR

Total RNA was extracted from cells using PureLink RNA Mini Kit (Invitrogen), and cDNA was synthesized using iScript cDNA Synthesis Kit (Bio-Rad). qRT–PCR was performed using iTaq Universal SYBR Green Supermix (Bio-Rad) on a CFX384 RT–PCR detection system (Bio-Rad).

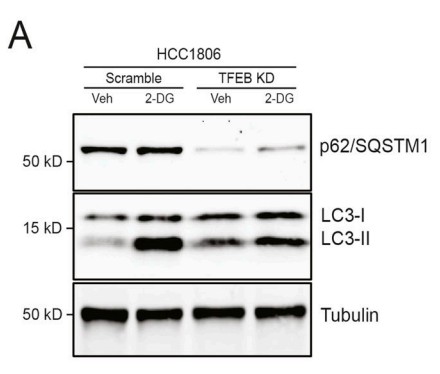

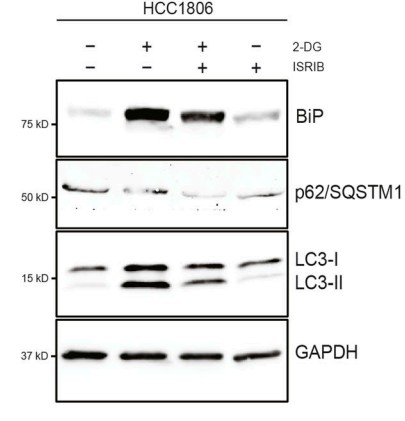

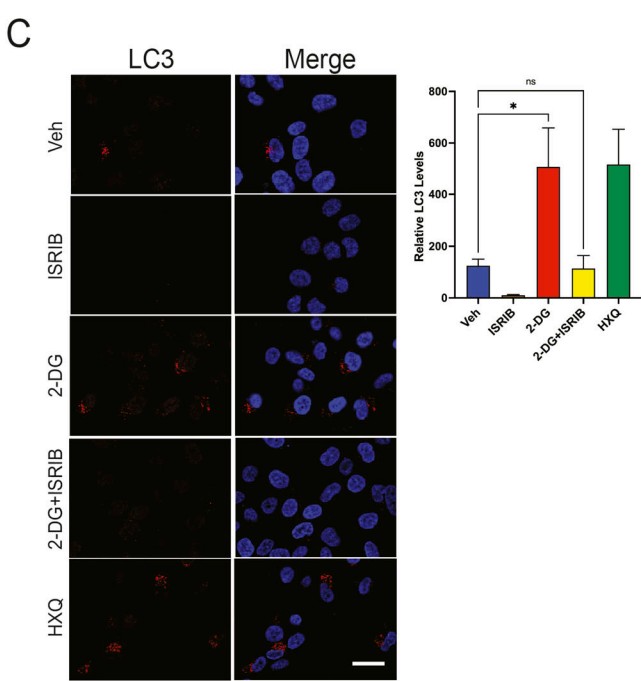

**Figure 7. TFEB induces autophagy in response to 2-DG.**
**(A)** Western blot of p62/SQSTM1 and LC3 in either scramble or TFEB KD treated with either vehicle or 2-DG for 24 h. **(B)** Western blot of LC3, p62/SQSTM1, and BiP in cells treated with vehicle, 2-DG, and/or ISRIB for 24 h. **(C)** Representative ICC images of LC3 and DAPI in HCC1806 cells treated with vehicle, 2-DG, ISRIB, 2-DG + ISRIB, or hydroxychloroquine (one-way ANOVA, Dunnett's test). Scale bar: 25 μm. Veh, vehicle; HXQ, hydroxychloroquine. *$P < 0.05$; **$P < 0.01$; ***$P < 0.001$; ****$P < 0.0001$.

## Mammosphere formation assay

The mammosphere culture medium was prepared with DMEM/F12 supplemented with GlutaMAX, penicillin/streptomycin, EGF, bFGF, and B27. Cells were trypsinized and plated in 24-well ultra-low attachment plates for 7–10 d. Primary mammospheres were trypsinized and replated in 24-well ultra-low attachment plates to form secondary mammospheres.

## Clonogenic assay

Cells were washed, trypsinized, and counted. Either 6-well or 12-well plates were seeded with $2.5 \times 10^3$ or $1.5 \times 10^3$ cells/well, respectively. Plates were incubated for 7–10 d at 37°C, 5% $CO_2$. Colonies were stained with crystal violet and imaged. Colonies were solubilized in 10% acetic acid to quantify clonogenic capacity and read for absorbance at 590 nm in a plate reader (BioTek).

## Gene overexpression and shRNA knockdown

Cells were subjected to gene knockdown using shRNAs targeting TFEB or HSPA5 via lentiviral transduction. A scramble shRNA was used as a control in all shRNA-mediated knockdown experiments. The overexpression of TFEB (S142A) and RagC (S75L) was carried out via lentiviral transduction. An empty pLenti-C-Myc-DDK-IRES-Puro vector was used as a control in all overexpression experiments.

## Limiting dilution tumor initiation assay and tumor xenograft assay

Female NSG (NOD.Cg-PrkdcSCID Il2rgtm1Wjl/SzJ) mice, purchased from the Jackson Laboratory, were orthotopically injected with HCC1806 cells expressing either scramble control or TFEB shRNA. Each cell line was injected with serial dilutions $5 \times 10^5$ (n = 5), $5 \times 10^4$ (n = 6), $5 \times 10^3$ (n = 7), or $5 \times 10^2$ (n = 9). Tumor growth in mice was

tracked for 140 d after injection. The tumor-initiating cell frequency for each cell line was calculated using ELDA (extreme limiting dilution analysis) (Hu & Smyth, 2009) at https://bioinf.wehi.edu.au/software/elda.

Female athymic (Foxn1$^{nu/nu}$) mice, purchased from Envigo, were orthotopically injected with $1 \times 10^6$ cells transduced with either scramble control or TFEB shRNA. Tumors were measured three times a week using a caliper, and tumor volume was computed as follows: (length×width$^2$/2).

## Statistical analysis

Statistical analyses were carried out using GraphPad Prism 9. The specific statistical tests used included $t$ test, one-way ANOVA followed by Dunnett's post hoc test, and two-way ANOVA followed by Sidak's post hoc test as denoted.

# Data Availability

The authors generated the data, which are available upon request.

# Supplementary Information

# Acknowledgements

This work was supported by CPRIT Grant RR160093 (to SG Eckhardt), CPRIT Grant RP210088 (to KN Dalby), UT College of Pharmacy Discretionary Funds (to C Van Den Berg), and UT Graduate Continuing Fellowship (to M Soleimani). We thank the DNA Sequencing Facility Core at UT Austin for Sanger sequencing, the Center for Biomedical Research Support at UT Austin for flow cytometry and microscopy, and all the Developmental Therapeutics Laboratory personnel for their guidance and support.

## Author Contributions

M Soleimani: conceptualization, data curation, formal analysis, supervision, methodology, and writing—original draft, review, and editing.
M Duchow: data curation, formal analysis, investigation, and methodology.
R Goyal: investigation.
A Somma: data curation and methodology.
TS Kaoud: investigation and methodology.
KN Dalby: conceptualization and methodology.
J Kowalski: formal analysis, supervision, and methodology.
SG Eckhardt: resources, formal analysis, and funding acquisition.
C Van Den Berg: conceptualization, supervision, data curation, formal analysis, funding acquisition, methodology, and writing—original draft, review, and editing.

## Conflict of Interest Statement

The authors declare that they have no conflict of interest.

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
