## [Reviewer comments · Life Science Alliance]

Life Science Alliance

Transcription Factor EB (TFEB) activity increases resistance of TNBC stem cells to metabolic stress

Milad Soleimani, Mark Duchow, Ria Goyal, Alex Somma, Tamer Kaoud, Kevin Dalby, Jeanne Kowalski, S. Eckhardt, and Carla Van Den Berg

DOI: <https://doi.org/10.26508/lsa.202302259>

Corresponding author(s): *Carla Van Den Berg, The University of Texas at Austin*

Review Timeline:

Submission Date:	2023-07-07
Editorial Decision:	2023-09-29
Revision Received:	2024-12-16
Editorial Decision:	2024-12-17
Revision Received:	2024-12-19
Accepted:	2024-12-20

Scientific Editor: *Eric Sawey, PhD*

Transaction Report:

September 29, 2023

Re: Life Science Alliance manuscript #LSA-2023-02259-T

Dr. Carla Lynn Van Den Berg
University of Texas, Austin
Pharmacy and Oncology at Dell Medical School
1601 Trinity Street, Building B
Austin, TX 78712

Dear Dr. Van Den Berg,

Thank you for submitting your manuscript entitled "Transcription Factor EB (TFEB) activity increases resistance of TNBC stem cells to metabolic stress" to Life Science Alliance. The manuscript was assessed by expert reviewers, whose comments are appended to this letter. We invite you to submit a revised manuscript addressing the Reviewer comments.

Thank you for this interesting contribution to Life Science Alliance. We are looking forward to receiving your revised manuscript.

Sincerely,

B. MANUSCRIPT ORGANIZATION AND FORMATTING:

Reviewer #1 (Comments to the Authors (Required)):

In this manuscript Soleimani et al., aim at investigating the metabolic effects of TFEB in breast cancer stem cells. Initially they aimed at showing that TFEB regulates breast cancer stem cell self-renewal. They also evaluated the effect of 2-DG, a glycolysis inhibitor, on TFEB as well as on UPR. In these studies, they used several cell lines and mammosphere formation as their functional assays. The authors have performed substantial amount of work but their studies have also some caveats as I describe below:

- 1) Figs 1-3 are confirmations of previously published results
- 2) Phosphorylation of TFEB should be evaluated in studies assessing TFEB activity in addition to subcellular localization; this is mostly missing throughout this study
- 3) In order to establish the effect of glycolysis on TFEB, the authors use only 2-DG, no other approaches are used which is insufficient
- 4) Fig 5B: It is stated that: "Stable overexpression of TFEB(S142A) lowered 2-DG sensitivity in TNBC mammospheres compared to control. In contrast, overexpression of RagC (S75L) enhanced 2-DG cytotoxicity" Is this quantified? I do not see differences in relative 2-DG effects in RagC (S75L) versus control. Please clarify.
- 5) Fig. 5B: The effect of TFEB (S142A) versus RagC (S75L) is significantly different on mammosphere formation, please comment. 2DG does not seem to have any effect in this case at the doses. Although I concur with the notion that TFEB may enhance self-renewal, the data as presented is not convincing. Is the difference seen due to the amount of protein expressed or an intrinsic effect of TFEB expression?
- 6) Fig. 5C: "A Western blot analysis showed that silencing TFEB blunted 2-DG upregulation of UPR markers " I don't see blunting the 2-DG effect on UPR, only mildly reducing some of the markers.
- 7) Figs 6 and 7 are preliminary.
- 8) The authors should clarify why they looked at UPR in this context.
- 9) Some of the interpretations of the results are overstated including in Fig 5C.

Reviewer #2 (Comments to the Authors (Required)):

This work addresses an important area of research i.e. cancer stem cells and their regulation by the MiT/TFE family of transcription factors. Here, TFEB was shown to be critical for mammosphere formation in vitro and tumor initiation/growth in vivo, with further evidence provided by a decline in CD44^{high}/CD24^{low} cells. Further evidence is provided showing that TFEB augments UPR-related survival. How certain points need clarification, possibly with additional experiments, as highlighted below:

- 1) Figure 1F - was endogenous TFEB not detected in the HCC1806 cells since it is expressed in these cells (Figure EV 1A). Also, was this experiment done in the other TNBC cells (MDA-MB-157 and MDA-MB-231). If not, this experiment should be performed to see if it increases mammosphere formation in all the TNBC cell lines.
- 2) Figure 3F- Dephosphorylation and a subsequent increase in nuclear localization of TFEB should be confirmed using validated site-specific phospho-TFEB antibodies like the ones below:
 - a) Phospho-TFEB (Ser211) (E9S8N) Rabbit mAb #37681
 - b) Phospho-TFEB (Ser122) (E9M5M) Rabbit mAb #87932

c) Anti-phospho TFEB (Ser142) ABE1971-I-100UL

Additionally, the effects of 2-DG on TFEB should be confirmed using nuclear-cytoplasmic fractionation immunoblots or immunofluorescence. The blots in EV3A are not confirmatory for TFEB and should be repeated.

3) Furthermore, if 2-DG is sufficient to induce TFEB nuclear localization and subsequent UPR, and if both TFEB nuclear localization and UPR are sufficient to promote CSC stemness, how do the authors explain the fact that 2-DG actually diminishes stemness? The effects of 2-DG on TFEB should be more thoroughly analyzed (as described in point 2 above). In addition, the effects of 2-DG on CLEAR promoter activity (4Xclear Luciferase reporter assays) and TFE3 subcellular localization (as a potential compensatory mechanism) should also be examined.

Reviewer #1 (Comments to the Authors (Required)):

In this manuscript Soleimani et al., aim at investigating the metabolic effects of TFEB in breast cancer stem cells. Initially they aimed at showing that TFEB regulates breast cancer stem cell self-renewal. They also evaluated the effect of 2-DG, a glycolysis inhibitor, on TFEB as well as on UPR. In these studies, they used several cell lines and mammosphere formation as their functional assays. The authors have performed substantial amount of work but their studies have also some caveats as I describe below:

1) Figs 1-3 are confirmations of previously published results

The data in Figures 1-3 are unique to this manuscript however, we did submit this manuscript to bioRxiv as suggested by Embo Molecular Medicine, the parent journal.

2) Phosphorylation of TFEB should be evaluated in studies assessing TFEB activity in addition to subcellular localization; this is mostly missing throughout this study

Fractionation studies are shown in Supplemental Figure EV3 which demonstrates that 2DG induces nuclear localization in parental HCC 1806 cells.

Fractionation studies are also shown in Figure 5A which demonstrates the preferential nuclear localization that 2DG induces in control HCC 1806 cells and the inhibition of nuclear location of TFEB when RagC(S75L) is overexpressed. We have also added new Western blots to Fig. EV3A showing inhibition of p-TFEB (S122) and p-TFEB (S211) in glucose-starved cells.

Finally, we performed both cytoplasmic and nuclear fractionation along with phospho-TFEB blotting to show that S122 is only phosphorylated in the cytosolic fraction, and that 2DG treatment is associated with the absence of S122 phosphorylation (figure shown here).

3) In order to establish the effect of glycolysis on TFEB, the authors use only 2-DG, no other approaches are used which is insufficient

We appreciate the reviewer bringing this to our attention. In order to make our observations more broadly applicable we have performed some additional studies using glucose starvation of TNBC cells. We have added data to figures EV3A and EV3C showing that glucose starvation inhibits the phosphorylation of TFEB on S122 and S211. Additionally, we show that lack of glucose increases AMPK mediated phosphorylation of ACC and inhibits the phosphorylation of 4EBP1-S65 (an mTORC1 substrate), supporting that like 2DG treatment, cells are experiences glycolytic stress. We have also performed a promoter reporter assay with 2-DG and glucose starvation, indicating TFEB activation of its transcriptional target, the CLEAR motif (Fig. 3F).

4) Fig 5B: It is stated that: "Stable overexpression of TFEB(S142A) lowered 2-DG sensitivity in TNBC mammospheres compared to control. In contrast, overexpression of RagC (S75L) enhanced 2-DG

cytotoxicity" Is this quantified? I do not see differences in relative 2-DG effects in RagC (S75L) versus control. Please clarify.

Thank you for the comment. The results were quantified and statistical analysis performed between vehicle and 2-DG treatment for each model, namely control, TFEB (S142A), and RagC (S75L). The effect of 2-DG was evaluated for each model with respect to the corresponding vehicle control. The experiment was repeated in two additional cell lines, HCC1937 and HCC38, where control and TFEB (S142A)-overexpressing cells were treated with either vehicle or 2-DG (Fig. EV3C). Similar to HCC1806, the new cell lines showed attenuated 2-DG response in the TFEB (S142A)-overexpressing model compared to control (Fig. EV3C).

5) Fig. 5B: The effect of TFEB (S142A) versus RagC (S75L) is significantly different on mammosphere formation, please comment. 2DG does not seem to have any effect in this case at the doses. Although I concur with the notion that TFEB may enhance self-renewal, the data as presented is not convincing. Is the difference seen due to the amount of protein expressed or an intrinsic effect of TFEB expression?

Indeed, TFEB (S142A) overexpression mitigates the toxic effect of 2-DG, while retaining TFEB in the cytoplasm by overexpressing RagC (S75L) has the opposite effect. The images show that TFEB (S142A) expression also causes larger mammospheres. The associated graphs do not show these inherent growth difference. Instead, we showed differences within each group, comparing 2DG to its relative vehicle control. Our conclusion is that TFEB activity allows mammospheres to tolerate 2DG induced glycolytic stress. Below we include comparisons of TFEB (S142A) versus control in absolute mammosphere counts using HCC1937 and HCC38 cells without treatment. In these experiments, expression of TFEB (S142A) significantly increased mammosphere numbers in both cell lines.

6) Fig. 5C: "A Western blot analysis showed that silencing TFEB blunted 2-DG upregulation of UPR markers " I don't see blunting the 2-DG effect on UPR, only mildly reducing some of the markers.

Thank you for the feedback. The blots have been quantified to better reflect the changes in UPR marker levels. The text has been modified as follows "A Western blot analysis showed that silencing TFEB decreased 2-DG upregulation of UPR markers CHOP, BiP, PERK, and IRE1 α ."

7) Figs 6 and 7 are preliminary.

Thank you for the opportunity to elaborate on the data in Figures 6 and 7. Multiple experiments and strategies were employed to validate the results in both figures. In Figure 6A, we confirm BiP knockdown and its effect on 2-DG-induced CHOP upregulation. In Figs. 6B, C, and D, we use three

different methods to validate the role of BiP in self-renewal and cancer stemness. The methods are chosen to assess the phenotypic implications of BiP KD as well as relevant CSC markers.

In Fig. 7, we use three different approaches to show that TFEB mediates the effect of 2-DG on autophagy.

In Fig. 7A, we show that TFEB KD mitigates the effect of 2-DG on autophagy markers. In Fig. 7B, we use a chemical approach to mimic the effect of TFEB KD. In Fig. 7C, we use immunofluorescence in order to validate the results in Fig. 7B.

8) The authors should clarify why they looked at UPR in this context.

To provide further clarity, additional literature and background were added. The following paragraph is now in the manuscript:

“Various components of UPR are closely associated with breast CSC character, suggesting that UPR preserves CSC populations. Metastatic breast tumors, with high levels of CD44^{high}/CD24^{low} cells, display an upregulation of Binding immunoglobulin Protein (BiP; a.k.a. GRP78, HSPA5) and Protein-Disulfide Isomerase (PDI) (Bartkowiak *et al*, 2010). Inhibition of ATF6 and PERK suppresses mammosphere formation (Li *et al*, 2018). Knockdown of X-box Binding Protein 1 (*XBP1*) reduces CD44^{high}/CD24^{low} enrichment in TNBC (Chen *et al*, 2014). Overexpression of BiP increases CD44^{high}/CD24^{low} cells and upregulates CSC-associated genes (Conner *et al*, 2020). PERK knockdown in mouse mammary carcinoma cells reduces tumor initiation and expansion (Bobrovnikova-Marjon *et al*, 2010).”

9) Some of the interpretations of the results are overstated including in Fig 5C.

Thank you for the feedback. The blots have been quantified to better reflect the changes in UPR marker levels. The text has been modified as follows “A Western blot analysis showed that silencing TFEB decreased 2-DG upregulation of UPR markers CHOP, BiP, PERK, and IRE1 α .”

Reviewer #2 (Comments to the Authors (Required)):

This work addresses an important area of research i.e. cancer stem cells and their regulation by the MiT/TFE family of transcription factors. Here, TFEB was shown to be critical for mammosphere formation in vitro and tumor initiation/growth in vivo, with further evidence provided by a decline in CD44^{high}/CD24^{low} cells. Further evidence is provided showing that TFEB augments UPR-related survival. How certain points need clarification, possibly with additional experiments, as highlighted below:

1) Figure 1F - was endogenous TFEB not detected in the HCC1806 cells since it is expressed in these cells (Figure EV 1A). Also, was this experiment done in the other TNBC cells (MDA-MB-157 and MDA-MB-231). If not, this experiment should be performed to see if it increases mammosphere formation in all the TNBC cell lines.

Thank you for the feedback. In Fig 1F we only used a myc antibody to detect overexpression of myc-TFEB so endogenous TFEB would not be detected in this blot. Figure EV1A is only showing endogenous TFEB, no overexpression of TFEB. In this figure, we used shTFEB expression to reduce

TFEB expression in three different cell lines. This is why strong TFEB bands are seen in the scrambled control groups of all three cell lines versus weak bands in the shTFEB groups.

We appreciate that the reviewer noted the lack of endogenous TFEB when we perform western blots using the TFEB mutant. When we use the TFEB antibody the higher expression of mutant TFEB does not always allow visualization of endogenous TFEB. An example is shown in Figure 5D. On the top blot we used different exposures for endogenous and overexpressed TFEB. On the bottom blot, endogenous and overexpressed mutant TFEB can be viewed using the same long exposure on the top panel. A shorter exposure only shows overexpressed TFEB in the second panel. We provide another example below that allows one to visualize both endogenous and overexpressed TFEB (S142A).

The mammosphere experiment has been repeated in two additional cell lines, HCC38 and HCC1937. In both cell lines, overexpressing TFEB (S142A) increased mammosphere formation. Also, TFEB (S142A) diminished 2-DG response compared to control in both cell lines (Fig. EV3C). Below, we also include comparisons of TFEB (S142A) versus control in absolute mammosphere counts using HCC1937 and HCC38 cells without treatment. In these experiments, expression of TFEB (S142A) significantly increased mammosphere numbers in both cell lines.

2) Figure 3F- Dephosphorylation and a subsequent increase in nuclear localization of TFEB should be confirmed using validated site-specific phosphor-TFEB antibodies like the ones below:

- a) Phospho-TFEB (Ser211) (E9S8N) Rabbit mAb #37681
- b) Phospho-TFEB (Ser122) (E9M5M) Rabbit mAb #87932
- c) Anti-phospho TFEB (Ser142) ABE1971-I-100UL

Additionally, the effects of 2-DG on TFEB should be confirmed using nuclear-cytoplasmic fractionation immunoblots or immunofluorescence. The blots in EV3A are not confirmatory for TFEB and should be repeated.

Thank you for the comment. Western blots showing p-TFEB (S122) and p-TFEB (S211) have been added to the manuscript (Fig. EV3A). Fractionation studies are shown in Supplemental Figure EV3 which demonstrates that 2DG induces nuclear localization in parental HCC 1806 cells. Fractionation studies are also shown in Figure 5A which demonstrates the preferential nuclear localization that 2DG induces in control HCC 1806 cells and the inhibition of nuclear location of TFEB when RagC(S75L) is overexpressed.

3) Furthermore, if 2-DG is sufficient to induce TFEB nuclear localization and subsequent UPR, and if both TFEB nuclear localization and UPR are sufficient to promote CSC stemness, how do the authors explain the fact that 2-DG actually diminishes stemness? The effects of 2-DG on TFEB should be more thoroughly analyzed (as described in point 2 above). In addition, the effects of 2-DG on CLEAR promoter activity (4Xclear Luciferase reporter assays) and TFE3 subcellular localization (as a potential compensatory mechanism) should also be examined.

Thank you for the insightful comment and the helpful suggestions. We have performed a promoter reporter assay using the recommended 4Xclear luciferase system in HCC1806 and HCC38 exposed to 2-DG, glucose starvation, and no treatment. Both cell lines showed an increase in CLEAR-Luc activity in response to either 2-DG or glucose starvation.

We looked at TFE3 levels in HCC1806 and HCC38 in response to glucose starvation for 3h, 8h, and 24h but did not see a consistent sumoylation as we have previously seen with JNK-IN-8. (Soleimani *et al.* 2022)

To address the relation between 2-DG, TFEB, UPR, and CSCs, we show that 2-DG causes glycolytic stress, which in turn activates TFEB and UPR. While it is true that 2-DG-induced glycolytic stress diminishes CSCs, it sets in motion a cell recovery mechanism mediated by TFEB. Although endogenous TFEB appears insufficient to rescue CSCs in response to glycolytic stress, overexpression of constitutively-active TFEB decreases the effect of 2-DG on CSCs. Conversely, knocking down TFEB, and UPR marker BiP by extension, reduces CSCs.

Soleimani M, Somma A, Kaoud T, Goyal R, Bustamante J, Wylie DC, Holay N, Looney A, Giri U, Triplett T et al (2022) Covalent JNK Inhibitor, JNK-IN-8, Suppresses Tumor Growth in Triple-Negative Breast Cancer by Activating TFEB- and TFE3-Mediated Lysosome Biogenesis and Autophagy. Mol Cancer Ther 21: 1547-1560.

December 17, 2024

RE: Life Science Alliance Manuscript #LSA-2023-02259-TR

Dr. Carla Lynn Van Den Berg
The University of Texas at Austin
Pharmacy and Oncology at Dell Medical School
1601 Trinity Street, Building B
Austin, TX 78712

Dear Dr. Van Den Berg,

Thank you for submitting your revised manuscript entitled "Transcription Factor EB (TFEB) activity increases resistance of TNBC stem cells to metabolic stress". We would be happy to publish your paper in Life Science Alliance pending final revisions necessary to meet our formatting guidelines.

- please be sure that the authorship listing and order is correct
- please add the Twitter handle of your host institute/organization as well as your own or/and one of the authors in our system
- please be sure that the authorship listing and order is correct
- LSA allows supplementary figures, but not EV Figures; please update your callouts for the Supplementary Figures in the manuscript Fig EV1A=Fig S1A; while supplementary figures use the system supplementary Fig S1
- please add an Author Contributions section to your main manuscript text
- the financial support section on the title page should instead be included in the Acknowledgments section
- please remove the Paper Explained section

LSA now encourages authors to provide a 30-60 second video where the study is briefly explained. We will use these videos on social media to promote the published paper and the presenting author (for examples, see <https://docs.google.com/document/d/1-UWCfbE4pGcDdcgzcmiuJI2XMBJnxKYeqRvLLrLS08s/edit?usp=sharing>). Corresponding or first-authors are welcome to submit the video. Please submit only one video per manuscript. The video can be emailed to contact@life-science-alliance.org

A. FINAL FILES:

B. MANUSCRIPT ORGANIZATION AND FORMATTING:

Sincerely,

December 20, 2024

RE: Life Science Alliance Manuscript #LSA-2023-02259-TRR

Dr. Carla Lynn Van Den Berg
The University of Texas at Austin
Pharmacy and Oncology at Dell Medical School
1601 Trinity Street, Building B
Austin, TX 78712

Dear Dr. Van Den Berg,

Thank you for submitting your Research Article entitled "Transcription Factor EB (TFEB) activity increases resistance of TNBC stem cells to metabolic stress". It is a pleasure to let you know that your manuscript is now accepted for publication in Life Science Alliance. Congratulations on this interesting work.

DISTRIBUTION OF MATERIALS:

Again, congratulations on a very nice paper. I hope you found the review process to be constructive and are pleased with how the manuscript was handled editorially. We look forward to future exciting submissions from your lab.

Sincerely,
